# Perception of Competence as Mediator between Motor Competence and Physical Activity

**DOI:** 10.3390/ijerph19010392

**Published:** 2021-12-30

**Authors:** Luis-de Cos Izaskun, Urrutia-Gutierrez Saioa, Luis-de Cos Gurutze, Arribas-Galarraga Silvia

**Affiliations:** 1Deparment of Musical, Visual Arts and Physical Education, Universidad del País Vasco, 48940 Leioa, Spain; izaskun.luis@ehu.eus (L.-d.C.I.); silvia.arribas@ehu.eus (A.-G.S.); 2Education Department, Universidad de Antofagasta, Antofagasta 1240000, Chile; gurutze.luisdecos@uantof.cl

**Keywords:** motor competence, perceived motor competence, physical activity

## Abstract

Background: The practice of physical activity (PA) plays an important role in achieving an active-healthy lifestyle. Several authors have focused their studies on the relationship between motor competence (MC) and physical activity (PA). Stodden et al. proposed a conceptual model, where they postulated the existence of a positive and significant relationship between the two variables and that there are mediating variables that play a critical role in this relationship as perceived motor competence (PMC). Aims: Therefore, the purpose of this research is to provide empirical evidence to support the conceptual model. The aim is to examine the association of PMC and MC with PA and to determine whether PMC is a mediator of the association between MC and PA in Basque adolescents. Methods: 897 students between 12 and 16 years old from the Basque Country (Spain) participated in this study. The SPORTCOMP battery was used to assess motor competence and the AMPET-R questionnaire was applied to measure the PMC. To know the level of PA participants were asked how many days per week they performed physical activity, considering physical activity to include any sport activity, organized or unorganized, that meets the conditions of a minimum duration of 60 min medium and vigorous intensity. A descriptive, correlational and direct/indirect effect approach was used, using the PROCESS macro for Statistical Package for the Social Sciences (SPSS). Results: The results highlight that, on the one hand, PMC significantly correlates with both MC and PA and, on the other hand, it is corroborated that PMC is a mediator variable in the relationship between MC and PA. Conclusion: The mediation role of the PMC in the association between MC and PA raises the necessity not only to improve motor skills but also to provide successful experiences that allow adolescents to build a competent image of themselves that will contribute to the achievement and maintenance of an active lifestyle.

## 1. Introduction

Today, the awareness of leading a healthy life from its broadest conception is becoming increasingly important. However, year after year the WHO warns of an increase in sedentary lifestyles among the general population and especially among adolescents [1]. One of the most important tools available to society to ensure that young people establish active-healthy habits that last into maturity is education; specifically, the subject of physical education plays a very important role. Physical education should encourage adherence to physical activity, thus helping students to be more active throughout their lives [2].

To intrinsically understand this research, it is necessary to focus the discourse based on the criteria established in the educational curriculum of the Autonomous Community of the Basque Country (Spain). The curriculum establishes as basic competence for the subject of physical education the motor competence which, as the educational curriculum of the Autonomous Community of the Basque Country indicates in Decree 236/2015, the motor competence [3]:

“It contributes to the development of all basic competencies through the improvement of physical abilities and skills, the use of motor behavior as a means of affirming one’s cultural identity and the values of the country, the use of the body as a means of communication and expression, and the integration of physical activity in daily life for the improvement of health.” (p. 147).

In the breakdown of the physical education curriculum, among the main objectives are established, on the one hand, to know and appreciate the benefits of the practice of physical and sports activities and, on the other hand, to provide a critical view on health and treatment of the body. Finally, to know how to plan autonomously and based on individual needs, recreational, expressive and sporting physical activities [3]. All these objectives are oriented to achieve an improvement in quality of life, giving priority to the achievement of healthy active habits through the practice of physical activity.

The practice of physical activity plays an important role in achieving an active-healthy lifestyle. Since learning to move is necessary to participate in physical activity [4], physical education has the responsibility of ensuring that children learn to move effectively. Research along these lines has focused on the relationship between motor competence (MC) and physical activity (PA). It has postulated the existence of a positive and significant relationship between the two variables [5,6,7]. In addition, Barnett et al. [8] showed that the development of MC is one of the predictors of the achievement of future practice habits [9].

In order to explain this relationship, Stodden et al. [4] proposed a conceptual- model establishing as a central axis the reciprocal and dynamic relationship between MC and PA. They suggest that MC is the main underlying mechanism that promotes commitment to physical and sports practice and that there are mediating variables that play a critical role in this relationship. Among others, they suggest the perception of motor competence (PMC). Different investigations [7,10,11,12] have exposed that the number of successes and failures accumulated during the experiences lived in an area, influence the creation of the PMC of each individual. Thus, the effective execution of motor tasks leads to a positive assessment of MC.

Stodden et al. [4] in their conceptual model, propose PMC as the mediating variable that influences the relationship between MC and PA. They add that these variables interact. Thus, those with low MC will show lower PMC and will find physical-sports practice less desirable, and this may even lead to abandonment. On the contrary, those who present moderate and high levels of MC will present higher PMC and will probably persist in physical and sport activities.

However, Stodden et al. [4] point out that this relationship could be a developmental phenomenon that changes over developmental time. In early childhood PMC is linked to attempts at mastery and task persistence; therefore, these authors expect that PMC is not strongly correlated with actual MC levels or PA. Studies by Hall, Eyre Oxford, and Duncan [13] and Lopes et al. [14] of early childhood boys and girls corroborate this hypothesis, finding no mediating effect of PMC on the association between MC and PA.

To provide empirical evidence that supports the conceptual model of Stodden et al. [4] and due to the limited number of studies that investigate PMC as a mediating variable in the association between MC and PA in adolescence, this research aims to explore the associations between MC, PMC and PA and to analyze the mediating effects of PMC in the associations between MC and PA in Basque adolescents (12 to 16 years old).

Confirming that PMC is a mediator in the association between MC and PA will allow us to orient physical education classes not only to the development and achievement of motor skills but also to propose activities that improve the feeling of competence which will influence engagement in physical-sports practice [4].

## 2. Materials and Methods

### 2.1. Participants

897 boys and girls participated in this study, 50.3% of them were boys and 49.7% were girls. All of them were resident in Basque Country (Spain) and aged between 12 and 15 years (M = 14.26; SD = 1.25) (Table 1).

### 2.2. Instrument

To measure motor competence, the SPORTCOMP battery by Ruiz, Graupera, García, Arruza, Palomo, Ramón [15], adapted by Arruza, Irazusta and Urrutia-Gutierrez [16] was used. It consisted of 10 different tests: five coordination capacity tasks and five physical condition capacity tasks. It is a battery of easy application on the one hand, because the material for data collection are accessible for the schools. On the other hand, because the number of tests is adapted to physical education lesson duration.

Coordination and motor control capacity. To quantify coordination capacity, the test battery consisted of five tasks: lateral jumps, 7 m jumping on one foot, 7 m jumping feet together, movement on support and balance. The performance measure was the time to completion in seconds for all tasks, except for the lateral jump task, which was the number of correct jumps performed.

Description:Lateral jumps: Lateral jumps with feet together on a board divided in half by a bar. As many jumps as possible in a time of 15 s.7 m hopping on one leg: Jumping on one leg over a distance of 7 m in the shortest possible time. Preferred leg shall be used.7 m jumping feet-together: Jumping with feet together over a distance of 7 m in the shortest possible time.Shift on supports: Moving over two supports at a distance of 3 m in the shortest possible time.Balance: One foot on a wooden bar, with hands on hips and eyes closed, keep balance without assistance.

Physical condition capacity. To quantify conditional capacity, the test battery consisted of five tasks: Medicine Ball Throwing, Flexibility, Dynamometry, Shuttle run and Abdominal. The performance measure was distance reached in centimeters for medicine ball throwing and flexibility, time to completion in seconds for the shuttle run, Newton for Dynamometry and Number of repetitions in 30 s for the abdominal test.

Description:Medicine Ball Throwing: Holding the ball with both hands at chest height, project it as far as possible.Flexibility: Flexibility box test.Dynamometry: With the extended arm, tighten the dynamometer.Shuttle run: In a marked space of 9 m, the student will run at maximum speed to pick up the first relay placed on the 9 m baseline and leave it behind the starting line. He/she will perform the same operation with a second relay. Once the relay baton has been placed on the ground after crossing the line, the test is over.Abdominal: Lying on the floor, with arms crossed over the chest and knees bent, sit up to 90°.

To measure the perceived motor competence, the scale extracted from the Achievement Motivation for learning in Physical Education (AMPET-R) by Tamotsu Nishida [17] and adapted to Spanish by Ruiz, Graupera, Gutierrez and Nishida [18], was used. The original version consists of 37 items and in this study; an internal version of 33 item was used. The answers were collected, indicating the degree of agreement/disagreement, on a Likert scale from 1 to 5 where 1 is complete disagreement and 5 is complete agreement. Internal consistency of the perceived motor competence was 0.92 according to Cronbach’s Alpha.

Physical Activity (PA) was assessed by asking the participants to indicate how many days per week they performed physical activity, considering physical activity to include any sport activity, organized or unorganized, that meets the conditions of a minimum duration of 60 min medium and vigorous intensity.

The biological sex was queried. Although sex in adolescence could be considered as a moderating variable of the mediation to be tested, the results showed that this was not a moderating variable; therefore, this variable was only considered to balance the sample.

### 2.3. Procedure

A trained assessor conducted the Sport-comp battery in physical education sessions that last two hours. After the physical test was completed, the questionnaires were complete. Consent forms and instructions for distributing these for children were delivered to participating schools. Consent was obtained from both schools and parents.

### 2.4. Data Analysis

Descriptive statistics were first calculated for all the variables under study (means and standard deviations). The internal consistency of each factor was analyzed using Cronbach’s Alpha. Pearson’s bivariate correlations were performed to analyze the relationships between the variables, macro PROCESS developed by Hayes [19] with a bootstrap threshold of 5000 and model 4 was used to performed the simple mediation analysis. We concluded that the indirect effect (IE) was statistically significant, if in the 95% confidence interval (CI) of the estimate zero was not included. Thus, whether PMC has mediation effect in the relationship between MC and PA is confirmed. The standardized (β) and unstandardized (B) regression coefficients are presented for four equations: (a) the equation that regressed the mediator (perceived motor competence) on the independent variable (motor competence), (b) the equation that regressed the dependent variable (Physical Activity) and the mediator, (c) the equation that regressed the independent variable and the dependent variable, and finally (c’) the equation that regressed the IE of the mediator (perceived motor competence) on the relationship between the independent variable (motor competence) and the dependent variable (Physical Activity).

## 3. Results

Table 2 reports la information related to the association between PA, MC and PMC. Bivariate correlations were carried out and the results revealed a positive and statistically significant correlation was observed between MC and PMC. PA was also positively and statistically significantly related to MC and PMC.

To determine whether PMC worked as a mediator variable between MC and PA, mediation analyses were carried out. The results showed that the effect of MC on PA was mediated by PMC (Figure 1). In the first regression equation (a), MC was positively related to PMC (β = 0.67; *p* = 0.000). In the second step (equation b) the regression coefficient of PMC on PA was significantly associated (β = 0.04; *p* = 0.000). In the last regression model (equation c), MC was positively associated with PA (β = 0.04; *p* = 0.000), but when PMC was included in the model (equation c’) the regression coefficient turned out to be non-significant and the relationship was eliminated (β = 0.03; *p* = 0.4855). Finally, the indirect effect was significant (indirect effect = 0.02) (95% CI; 0.02, 0.03), confirming the mediation role of PMC in this model.

## 4. Discussion

The present study examined the association of MC and PMC with PA during adolescence in a group of young Basques. From this point of view, the data of our research have confirmed this statement, showing on the one hand that a higher level of MC is associated with higher levels of PA and on the other hand, that PMC mediates the relationship between MC and PA.

Increasing age is associated with a decrease in PA [20,21,22,23,24,25] and MC may be a modifiable factor that can increase PA levels [26]. The results of the present study suggest that more competent adolescents present higher levels of PA and are more likely to continue practicing in PA than those with lower levels of MC [6,27]. The results obtained in this study support the findings of other studies [6,12,27,28], in which the positive association between PA and MC was confirmed. This association was supported by Barnett et al. [8], who proved the predictive value of MC in explaining PA, reinforcing the idea of the impact of CM in the maintenance and increment of PA.

Furthering this relationship, Stodden et al. [4] propose that there are additional factors that interact in the relationship between MC and PA, such as, for example, the PMC. The data from this study confirm that there is a positive and significant relationship between these three variables. On the one hand, following the tendency of previous research [6,12,29], the adolescents in this study with the highest levels of PMC are also those with the highest levels of PA. On the other hand, together with García-Canto et al. [30], Gómez [31], Gu et al. [32], Mata [33], Piek et al. [34], Vedul-Kjelsås et al. [35], our results allow us to affirm that students who present higher levels of MC perceive themselves as more competent, while those who manifest lower levels of MC perceive themselves as less competent. These results support the idea of McIntyre [36] and Barnett et al. [8], who propose MC and PMC as driving variables in the PA of adolescents.

In order to understand the relationship between the three variables, this research is based on the conceptual model of Stodden et al. [4], which allows us to consider the PMC as a mediating variable in the relationship between MC and PA.

Our mediation analysis confirms this idea, showing that PMC mediates the relationship between CM and PA, eliminating this relationship. This means that the direct effect of CM on PA disappears when PMC is included in the model, generating an indirect effect of MC on PA through PMC. Research by Barnett et al. [37], Gu et al. [32] and Khodaverdi et al. [12], as in this study, confirm this mediating effect.

In order to explain this process, Stodden et al. [4] suggest that the notion of MC is an underlying mechanism of PA. They state that MC interacts with the PMC, having great influence on the maintenance and persistence of PA. They consider that the presence of low levels of MC will be significantly related to low PMC and, subsequently, to low levels of PA. Barnett et al. [37] support this idea and indicate that being able to perform motor skills competently in childhood can be significant and influential in the construction of a positive PMC, which subsequently increases adolescents’ commitment to PA. 

However, Hall, Eyre, Oxford and Duncan [13] in a study conducted with a sample of British children in early childhood, did not obtain the mediating effect of the PMC in the association between MC and PA. These authors argued that the lack of the mediating effect was due to the early age of the sample, stating that the mediating effects of PMC happens in middle or late childhood and not in early childhood. Lopes et al. [14] obtained similar results with a sample of Portuguese children aged 5 to 9 years, concluding that there was poor association between MC and PMC.

These studies provide support to the idea that the mediating effects of PMC on the association between MC and PA is age-related. In adolescence, children have moved to higher levels of cognitive development and have more sophisticated cognitive ability to more accurately assess their MC, since, at this age, PMC is an indirect measure of actual MC [4].

That is why it is so important at school age to work and develop a good MC so that perceptions about it increase. This will have an impact on a greater practice, since authors such as Stodden et al. [4] and Welk [38], postulated that PMC is one of the most powerful mechanisms that influence the commitment and persistence in PA.

Therefore, based on our results, we propose to promote actions to increase the level of MC, which at the same time will influence on the PMC and will ensure the continuity of PA in adolescents. This idea is supported by the results obtained by Barnett et al. [8], where in their longitudinal study they corroborated that MC in childhood was a predictor of PA in adolescence. Taking into account different studies [24,39,40] that corroborate the idea that sports habits are established throughout adolescence, and that Kjonniksen et al. [41] affirm that PA in adolescence lasts into adulthood, we consider that children who participate in PA and reach higher levels of MC during childhood and adolescence will continue being active practitioners of PA into adulthood.

This study makes a novel contribution to the current literature. On the one hand, it provides empirical evidence to the conceptual- model of Stodden et al. [4] confirming this model in Basque adolescents. On the other hand, the results extend previous research by Barnett et al. [37], Khodaverdiet al. [12], Gu et al. [32], Lopes et al. [14], Hall et al. [13], acknowledging that PMC mediates the association between MC and PA in adolescence.

Therefore, in order to meet the requirements of the educational curriculum, to develop motor competence and to acquire healthy physical habits, physical education must propose appropriate and diverse activities. It should guarantee equal opportunities and should emphasize inclusive and varied activities in content.

Furthermore, this study proposes the application of cooperative learning. In a comparative study, Fernandez-Rios [42] showed that students who participated in physical education sessions through cooperative learning perceived themselves to be more skilled than those who worked with traditional methodology. As stated by Velázquez [43], improving the perception of motor competence while working in a team will allow an improvement in class relationships and this will contribute to increase the motivation of students towards physical education classes and eventually towards the practice of physical activity.

On the other hand, the increase in physical education hours will allow students to have the possibility of a greater participation in physical activity and thus have greater opportunities to increase the level of MC. In addition, this participation itself will help to develop higher PMC [4]. Malina, in 1996 [44] already postulated that individuals who during childhood and adolescence reach higher levels of MC will continue to be active in adulthood.

This research has some limitations. Firstly, the cross-sectional nature of the study does not allow us to assemble that this association is derived from a causal relationship. Secondly, the use of a self-report measure to assess PA may have raised some degree of recall bias and desirability.

## 5. Conclusions

This study aimed to explore the associations between PA, MC and PMC, and to evaluate the mediating effects of PMC between PA and MC in Basque adolescents. This study empirically confirmed the conceptual model of Stodden et al. [4], where perceived MC had mediating effects on actual MC and PA.

Considering the mediating influence of perceived CM on the association between motor competence and sport practice in adolescence, physical education has the responsibility to improve the current motor competence of students, as well as to propose activities that increase the perception of competence, since it will report an increment in the levels of sport practice.

These findings lead us to conclude improving and developing motor skills and abilities and further empowering students and providing successful experiences that allow them to build a competent image of themselves contributes to the achievement and maintenance of an active lifestyle.

Future research could perform longitudinal analyses in order to deepen these associations and understand how they evolve into adulthood.

## Figures and Tables

**Figure 1 ijerph-19-00392-f001:**
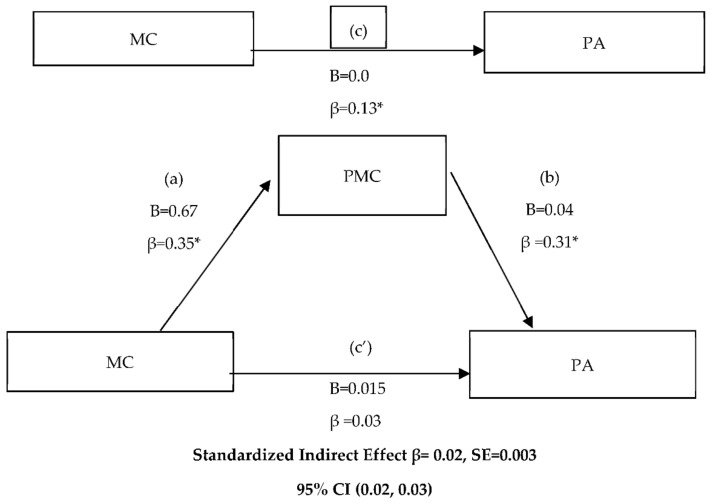
Simple mediation analysis of motor competence on the physical activity in relation to perceived motor competence. (**a**) the equation that regressed perceived motor competence on the motor competence, (**b**) the equation that regressed physical activity and perceived motor competence, (**c**) the equation that regressed motor competence and physical activity, and finally (**c’**) the equation that regressed the IE of perceived motor competence on the relationship between motor competence and Physical Activity. B: Unstandardized regression coefficient; β: Standardized regression coefficients; SE: Standard error, CI: Confidence interval. * Statistically significant at *p* = 0.000.

**Table 1 ijerph-19-00392-t001:** Descriptive data of the sample (N = 897).

Variable	Value	Number	Percentage
Sex	Boys	451	50.3
	Girls	446	49.7
Age	12	226	23.1
	13	230	23.5
	14	262	26.8
	15	259	26.5

**Table 2 ijerph-19-00392-t002:** Descriptive and correlational analyses between physical activity, motor competence and perceived motor competence.

			Correlations	
	M	SD	PA	MC
1. Physical Activity	49.92	9.63	-	-
2. Motor competence	50.02	5.05	0.133 **	-
3. Perceived motor competence	6.84	1.24	0.322 **	0.351 **

Note: ** *p* < 0.001; M = Mean; SD = Standard Deviation.

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
