# Peer review of "Perception of Competence as Mediator between Motor Competence and Physical Activity"

_ijerph, 2021, doi:10.3390/ijerph19010392_

Round 1

Reviewer 1 Report

This study investigated whether the perception of competence is a mediating variable in the relationship between motor competence and physical activity. Although the analysis processes followed the standard procedure, the rationale of this study is not clear. This paper contains many long sentences which makes it very hard to read. The author should make sentences concise. The structure of this paper should also be improved before publication.

Introduction

The rationale of this study is not clear. The author indicated that Stodden et al. has shown the perception of competence is a mediating variable in the relationship between motor competence and physical activity. The reason to re-conduct a similar study is not clear. Furthermore, the advantage or effect that identify whether the perception of competence is a mediating variable should be addressed. Long sentences should be broken into concise illustrations to improve readability.

Material and Methods

L89 What are M and S stand for?

The reason to choose the instrument should be addressed. Why use SPORTCOMP battery instead of others?

Results

Table 1 Correlation 1 and 2 is not clear? What are they stand for?

Discussion

Quantitative results should be illustrated and compared with previous literature (e.g. Stodden et al.), especially since the correlation coefficient is not high.

The effect of identifying the perception of competence as a mediating variable should be highlighted. Suggestions for adapting the curriculum should be also emphasised.

Conclusions

This should be in the discussion section. A new summary should be added in the conclusion sections.

Reviewer 2 Report

Congratulations on the contribution. The instrument used is original and provides highly relevant data for the investigation.
However, the age of the participants is a vital moment at the level of physical activity, so this article does not make sense if the data is not disaggregated by gender. Furthermore, the discussion should also provide this insight.

It is recommended to perform the statistical analysis to know if the data are different by gender.

If there are significant differences, you must present the results disaggregated by sex.

In the discussion section, reference should also be made to the difference  

Author Response

First of all thank you for you review.

Related to the disaggregation of sex, we did not found difference between boys and girls. We also did moderated mediation analysis to see if the sex was a variable that could moderate this mediation.

The results showed that sex did not moderate this relation. Our conclusion was that despite the fact that girls may perform less physical sports practice and have a lower level of motor competence, the perception that adolescents have of their abilities influences in the same way regardless of whether they have a higher or lower level of competence or whether they are boys or girls.

Reviewer 3 Report

TITLE: IMPROVE

SUMMARY: IMPROVE. The MET, the RESULTS and the CONCLUSION are not clearly indicated.

INTRODUCTION: Complete with more current research.

MATERIALS AND MEANS: Participants: Specify more data from them participants. It is recommended to make a table with all the data of the participants. Instrument: Better describe the instrument. Procedure: The procedure has not been accurately described.

RESULTS: Several tables should be included to show the results obtained.

DISCUSSION AND CONCLUSIONS: The conclusions obtained related to the objectives and results of the research should be completed and clearly offered

The manuscript must be reviewed in its entirety in terms of its wording and content.

Author Response

  1. REVIEWER

TITLE:

  • The title has been changed.

ABSTRACT:

  • The abstract has been improved.

INTRODUCTION

  • New references have been introduced.

PARTICIPANTES:

  • We had considered the recommendation and we presented a table with the data of the participants.
  • Taking into account Reviewer 1 suggestions, the description of the instrument has been changed.

RESULTS:

The results obtained are exposed in the table 1 and the figure 1. For the exposition of the mediation results we have followed the structure used for this type of analysis in different articles published iin MDPI journals. (e.g.: doi:10.3390/sports7040077; doi:10.3390/ijerph17072350; doi:10.3390/ijerph17238800 )

DISCUSSION AND CONCLUSIONS: 

  • Discussion and conclusions have been changed to complete and to offer clearly the information.

The manuscript has been reviewed in terms of wording and content.

Round 2

Reviewer 2 Report

Thanks for the clarification, but you have to clarify it also in the manuscript. It is necessary to provide this information in the methodology, results and discussion section.

Author Response

Gracias por tu reseña. Se ha aclarado en el documento.

Reviewer 3 Report

The article has been improved with the indications that we provide. It can be published with small spelling corrections. It is advised that you review the entire article an expert in English.

Author Response

Gracias por tu reseña. Hubo algunos errores ortográficos que se han corregido.